# Demographic Influences on Perceived Stressors of Construction Workers during the COVID-19 Pandemic

**DOI:** 10.3390/ijerph19074192

**Published:** 2022-04-01

**Authors:** Huakang Liang, Wenqian Yang, Tianhong Liu, Fan Xia

**Affiliations:** School of Economics and Management, Beijing Jiaotong University, Beijing 100044, China; 19241152@bjtu.edu.cn (W.Y.); 19241192@bjtu.edu.cn (T.L.); 19241045@bjtu.edu.cn (F.X.)

**Keywords:** demographic influences, COVID-19, perceived stressors, occupational health, construction workers

## Abstract

Construction work is one of the most stressful occupations in the world, and the COVID-19 pandemic has only exacerbated this reality. This research conducted a detailed investigation on the perceived stressors of different demographic groups among construction workers. Empirical data were collected using a structured questionnaire in the Chinese construction industry. The empirical data were processed using both an independent sample *t*-test and an Analysis of Variance (ANOVA). The findings indicated that male workers reported greater workloads than did females. Married workers experienced more pandemic fear and job insecurity, and they were more sensitive to the poor working environment. Highly-educated workers were inclined to be more satisfied with organizational pandemic responses, and had lower job insecurity and role ambiguity, but they experienced heavier workloads. In addition, the differences in work experience and age were statistically significant with regards to job insecurity. This research contributes to the body of knowledge by giving a comprehensive understanding of demographic influences on perceived stressors among construction workers. It also provides valuable insights to identify sensitive demographic groups and promote their health and wellbeing during and after the pandemic.

## 1. Introduction

In the beginning of 2020, the coronavirus disease (COVID-19) swept the globe, after which the World Health Organization (WHO) declared a global pandemic on 11 March 2020 [1]. As social distancing remains the primary strategy to minimize pandemic spread, the current COVID-19 pandemic has significantly affected economic activities involving human interactions [2]. As a typical labor-intensive industry, the construction industry is especially vulnerable to the COVID-19 crisis because remote working cannot replace physical construction activities onsite where construction workers are going to interact with each other and risk getting COVID-19 [2,3]. It is not surprising that the construction industry has suffered from one of the highest COVID-19 infection rates during the pandemic [4,5]. In addition, unprecedented economic impacts have also been caused by the COVID-19 pandemic through suspending and cancelling projects under development, and by interrupting the supply chain and the shortage of workers due to quarantines [3,6]. For instance, the average unemployment rate of construction workers in the USA increased by 95%, from 4.5% in 2019 to 8.7% in 2020 [7]. Therefore, COVID-19 has introduced significant challenges for frontline construction workers, such as increased concerns about economic insecurities and wellbeing, as well as the implementation of new and changing guidances in order to reduce the spread of the virus [8].

There have been several COVID-19-related studies specific to the construction industry. Some studies were focused on the impact of COVID-19 on the industry with regard to cost overruns and schedule delays [6,9,10,11,12]. For instance, Al-Mhdawi et al. (2020) identified the impact of COVID-19 on construction projects in Iraq [10], while Jeon et al. (2022) proposed the Purdue Index to assess the impact of COVID-19 on the US construction industry [6]. Other studies explored the response strategies on pandemic impacts [13,14,15,16,17,18]. For example, Zamani et al. (2022) provided pandemic response strategies in the construction industry for the government [13], while Kim et al. (2021) proposed an approach to examining the feasibility of pandemic guidelines implemented on construction sites in South Korea [14]. However, these studies primarily focused on the economic, financial, and operational consequences of the COVID-19 pandemic, such as reduced productivity and disruptions in the supply chain, while the research on workers occupational health and safety during the pandemic is still lacking, and also given that we know little about how construction workers perceive their work stressors. 

Previous studies have argued that construction is one of the most stressful industries, even prior to the COVID-19 pandemic [19]. For instance, construction workers are 1.7 times more likely than workers in general industries to suffer from psychological health issues, including emotional and stress-related injuries [20]. Stress is mainly due to various aggravators such as the harsh outdoor environment, physically demanding tasks, time pressure, interpersonal conflicts and insufficient management support [21]. The COVID-19 pandemic may further deteriorate this situation by disrupting the construction activities, and bringing about other stressors such as pandemic fear and job insecurity. Working under excessive stress could have adverse effects such as higher mistakes and accident risks, lowered productivity and mental health illnesses [21]. The construction industry is labor-intensive, and workers are directly responsible for the success of construction projects [22]. Therefore, it is imperative to investigate their perceptions with regard to the different stressors during the pandemic to identify potential opportunities to maintain their wellbeing and work performance [23].

The extent to which construction workers perceive the workplace stressors during the pandemic may vary among individuals [24]. Individuals with certain demographic characteristics may be more likely to feel a higher risk of infection, greater job insecurity, and increased work demands than other workers. For instance, in general perspective crossing industries, married workers were more likely to worry about job insecurity induced by the COVID-19 pandemic [25]. Therefore, if the demographic characteristics related to perceived stressors among construction workers can be identified, health and safety interventions for specific groups of construction workers can be carried out sufficiently [26]. Previous demographic studies on construction workers have explored the demographic influences on safety behaviors [24,27] and other safety-related cognitive activities such as safety perception [28] and representativeness heuristics in decision-making [26]. However, there is still a lack of studies on the impact of demographic or subgroup factors on construction workers’ perceptions of stressors, especially during the current pandemic. To address this limitation, the present study will examine workers’ perceptions of stressors among various demographic groups during the COVID-19 pandemic. 

This research will contribute to the body of knowledge on construction workers’ job stressors during the pandemic by investigating the effects of a comprehensive list of demographic factors (e.g., gender, marital status, education, age and working experience). It gives new insight into why some groups of workers exhibit poor health status on construction sites. This would help to assist managers to identify those construction workers most at risk and target this high-risk group with limited prevention resources.

## 2. Literature Review

### 2.1. Perceived Stressors

Construction work has been regarded as one of the most stressful occupations because the workers face physically demanding tasks, time pressure, interpersonal conflicts and poor working environments [22]. They are affected by various work-induced stressors, among which role stressors and poor working environment are two that are commonly investigated [29,30]. Role stressors emerge when individuals feel that they cannot properly understand and fulfill the role’s expectations and responsibilities, which are manifested by role overload, role conflict and role ambiguity [31]. By contrast, a poor working environment represents the physical sources of stress inherent in construction projects, such as the temperature, noise and safety hazards onsite. 

Furthermore, some unique psychological effects were induced by the current pandemic, mainly including the fear of infection, organizational pandemic responses, and job insecurity [32,33]. Pandemic fear represents the individual perceived risks of being infected by COVID-19 as well as other health consequences (e.g., the high mortality rate). Job insecurity reflects individual concerns about the prospect of job continuity due to the sluggish economy that has been triggered by this pandemic. Organizational pandemic responses capture whether organizational responses are adequate to protect workers from the current pandemic. 

#### 2.1.1. Pandemic Fear

One psychological response commonly reported is fear towards COVID-19 infection [34,35]. The COIVD-19 pandemic has caused exceptionally high levels of fear due to its worldwide spread, high media attention, lack of public knowledge and effective medical treatment, and drastic and unprecedented preventive measures (e.g., lockdowns and quarantines [36]). The fears related to the COVID-19 pandemic include not just fears of contracting the virus and worries about family members’ health but also the possible adverse economic outcomes due to the lockdowns [37,38]. Nevertheless, fear may be beneficial, as it motivates preventive behaviors such as hand-washing and social distancing [39]. However, fear can become maladaptive when it is excessive, leading to significant levels of distress and irrational behaviors. As such, it is important to investigate the fear of COVID-19 among construction workers to ensure that it is well-managed so that the detrimental consequences resulting from excessive fear can be minimized. 

#### 2.1.2. Job Insecurity

It is well-documented that the COVID-19 pandemic has caused sharply rising unemployment rates globally together with increasing concern about job insecurity among the workforce [40]. The construction industry is no exception. Many construction projects have been halted or postponed due to an interrupted supply chain and employee shortages due to quarantines [41]. Such changes could threaten workers and evoke a sense of job insecurity, which represents the perception of a potential threat to continuity in their current jobs [42]. Furthermore, most construction workers, especially in developing countries, come from poor socio-economic backgrounds [43]. Thus, they would experience serious financial hardships during the pandemic and therefore pay more attention to the stability of their jobs. 

#### 2.1.3. Organizational Pandemic Responses

The construction industry is highly susceptible to COVID-19 due to the physical presence of a large on-site workforce within a limited space [44]. As the world emerges from lockdown, construction remains a high-risk industry due to the threat of a second wave, where the organizations in this industry must learn how to conduct business while remaining safe at the same time. During COVID-19, when complying with safety measures such as the wearing of protective equipment and the enforcement of environmental hygiene, the organization could not only protect the health and safety of frontline staff, but also signal that the organization has a positive orientation towards their health and well-being. When they view the organization’s COVID-19 responses in a negative manner, workers are less likely to trust the organization and feel confident and encouraged to contribute to their work. 

#### 2.1.4. Role Stressors

Role overload is particularly salient to blue-collar construction workers [45]. Too demanding tasks may be mismatched with the ability of individual workers, which can lead to physical fatigue, lower productivity and more mistakes [46]. Role conflict occurs when the workers are subjected to inconsistent expectations as a result of conflicting demands such as the incompatibility between safety and productivity. In addition, a construction project normally involves many uncertain tasks due to the vague expectations of clients, insufficient substructure information, the dynamic economic situation, and the complicated organization structure, which potentially leads to role ambiguity. Role ambiguity takes place when there is insufficient information to understand and interpret role expectations. Both role conflict and role ambiguity will drive individuals into an ambiguous position where they are unsure how to operate correctly, thus impairing work performance [47]. 

#### 2.1.5. Poor Working Environment

Construction workers often have to work in poor physical conditions, such as extremely high or low temperatures, with inappropriate lighting, excessive noise and air pollution [48]. Long-term work in the harsh environment of construction sites not only damages individual health but also decreases their ability to perform tasks [30]. In addition, working on construction sites is inherently dangerous because of outdoor operations, working at heights, complicated on-site plants and equipment operation [49]. An unsafe working environment could increase individual stress and lead to costly accidents. 

### 2.2. Demographic Factors

Individuals from different demographic backgrounds may perceive various stressors during this pandemic in different ways. For the COVID-induced stressors, female workers have been found to report higher levels of fear of becoming infected, anxiety, and depression during the pandemic [50,51,52]. Elder people reported their greater worries about the severity of getting COVID-19 than the younger [53]. Afshari et al.’s (2021) investigations found that age, work experience and the level of education of frontline workers could predict their positive perceptions and responses towards the COVID-19 pandemic [54]. Narayana et al. (2020) argued that respondents aged more than 40 years with high education levels were positively associated with better perceptions towards COVID-19 [55]. In addition, marital status was found to increase the level of awareness of psychological health and physical activities to reduce the stress, while married workers were more concerned about job insecurity [56].

Regarding work-induced stressors, some studies have also suggested that demographic factors could have an effect on workplace perception among construction workers. For instance, Loosemore and Waters (2004) found that males experienced slightly higher levels of stress than women in the construction industry (Loosemore and Waters [57]). Wang et al. (2018) reported that age and working experience were negatively related to safety-related stressors (e.g., role ambiguity and role conflict) perceived by construction workers [47]. In addition, young workers were found to have more difficulties in dealing with stress due to lower levels of experience in handling job demands [58]. Maqsoom et al.’s (2020) investigation indicated that younger construction workers were more susceptible to work environmental stressors such as the inaccessibility to effective tools and the high outdoor temperatures, while less experienced workers were more concerned about regional economic and politic stressors [59]. Less educated workers tended to be much more risk-tolerable and be at higher risk of accidents than these highly-educated workers [60]. Following the literature review, we propose that construction workers suffer different stressors based on their demographic characteristics. Hypotheses about demographic influences on perceived stressors of construction workers during the COVID-19 pandemic were established:

**Hypothesis** **H1.**
*Gender difference has a significant influence on perceived stressors.*


**Hypothesis** **H2.**
*Marital status difference has a significant influence on perceived stressors.*


**Hypothesis** **H3.**
*Education level difference has a significant influence on perceived stressors.*


**Hypothesis** **H4.**
*Working experience difference has a significant influence on perceived stressors.*


**Hypothesis** **H5.**
*Age*
*difference*
*has*
*a*
*significant*
*influence*
*on*
*perceived*
*stressors.*


## 3. Method

### 3.1. Measures

A questionnaire was specially designed for this study to collect data related to perceived stressors among construction workers during the COVID-19 pandemic. Based on their experience during the COVID-19 pandemic, respondents were asked to rate their levels of agreement with the statements given on a five point Likert scale, a scale ranging from 1 (strongly disagree) to 5 (strongly agree). Structured questions of all constructs were adapted from self-report items validated by previous studies. These scales should be widely-used and be validated in similar context about workers’ occupational health, especially under the current COVID-19 pandemic. Prior to formal administration, a pilot study was performed to ensure that all items were appropriate for capturing the data necessary for this research. A total of ten workers participated in this pilot study, and based on it, the initial questionnaire was modified slightly (such as simplifying the language to make it clearer for workers with limited education). In addition to five demographic questions, namely gender, marital status, age, education level, and working experience, the final questionnaire included twenty-two specific questions to investigate workers’ perceived stressors during the pandemic, which have been provided in Appendix A.

#### 3.1.1. Pandemic Fear

The five questions for assessing the level of fear of COVID-19 among construction workers were adapted from the study of Chi, et al. [61] Sample items include “I believe that COVID-19 has a high fatality rate” and “I am worried about myself, my family members or my colleagues who may be affected by COVID-19”.

#### 3.1.2. Job Insecurity

Job insecurity was assessed by using three items which were constructed based on the study of Tang, et al. [62]. These items measured how insecure construction workers felt about their jobs. Sample items include “The COVID-19 pandemic makes me feel that my work is unstable” and “The COVID-19 pandemic makes me feel that my job prospects will change”.

#### 3.1.3. Organizational Pandemic Responses

Organizational pandemic responses were assessed by using three items, asking questions regarding construction workers’ satisfaction on the pandemic measures taken by the organizations [63]. Sample items include “I am satisfied with the way that my project responded to COVID-19” and “My project took care of its workers’ needs resulting from COVID-19”.

#### 3.1.4. Role Overload

Role overload was measured with three items that describe construction workers’ perception of how busy they are on a daily basis [64,65]. Sample items include “I often fail to finish my work on time.” and “My work requires a lot of time and effort.”

#### 3.1.5. Role Ambiguity

Four questions were asked to measure the construction workers’ unclear perception of work instructions and work content [66]. Sample items include “I do not exactly know what is expected of me” and “There is no clear, planned goals and objectives for my job”.

#### 3.1.6. Role Conflict

Construction workers’ sense of job conflict was assessed by four questions [66]. Sample items include “I have to buck a rule or policy in order to carry out an assignment” and “I receive incompatible requests from two or more people”.

#### 3.1.7. Poor Working Environment

Four questions were asked to measure the severity of working environment in terms of temperature, noise, air quality and risk [30]. Sample items include “My working environment has poor air quality, such as dust and heavy smoke” and “My working environment is at high risk”.

### 3.2. Questionnaire Survey

Between January and April of 2021, 800 questionnaires were handed out to construction workers on 21 different construction sites located in three provinces in China. These projects were selected based on the previous working relationships as well as the permission of the corporate level managers, and 30–50 workers were randomly recruited from each project. Construction workers were assured of anonymity and that their participation would be voluntary. After removing incomplete responses, a total of 498 valid responses were received (a response rate of 62.3%). The vast majority of the participants were male (81.7%) (shown as Table 1). Most respondents were married (82.7%). In terms of age, around 41.4% of the respondents were aged between 41 and 50, 32.7% were aged between 31 and 40, 17.1% were aged between 21 and 30, 8.4% were aged between 51 and 60, and 0.4% were under 20 years of age. The education level of the respondents was relatively low, with 84.8% having merely completed senior high school or less. The respondents were generally experienced construction workers, with more than 70% of them having over five years of work experience in the construction industry.

The distribution of respondents could meet the general nature of Chinese construction workers. The demographic characteristics involved in this research, as mentioned above, has also been reported consistently by other studies that were conducted in Chinese construction industry recently (shown as Table 1) [67,68,69]. Specifically, the construction industry has a consistently low amount of female workers in China, as about 80% are male. The majority are married and are aged between 30 and 50. Most workers had a high school education or less, indicating that the education level of Chinese construction workers is still far from ideal. Most workers have been working in the construction industry for more than five years, indicating that the Chinese construction industry has attracted a relatively stable group of migrant workers in past years.

### 3.3. Data Analysis

The collected data was performed using Analysis of Moment Structures (AMOS) v21.0 and SPSS v18.0 (IBM, Armonk, NY, USA). The questionnaire data was then assessed for reliability and validity to test the effectiveness of the scales. Convergent validity was analyzed through composite reliability, factor loading indicators, and Cronbach’s coefficients. Discriminant validity was verified if the square root of the average variance extracted (AVE) for a factor was greater than its largest inter-construct correlation [70]. Subsequently, the significance of any differences for descriptive comparisons was examined through an independent sample *t*-test and ANOVA (Analysis of Variance). To meet the requirements of ANOVA, the score of each construct was determined by calculating the mean value of its corresponding items [71].

## 4. Results

### 4.1. Descriptive Analysis

Descriptive statistics of all variables were presented in Table 2. The values of skewness and kurtosis of the constructs were all within the range of ±2.0, indicating that the data could better meet the normal distribution [72]. Generally, respondents rated organizational response to COVID-19 and job insecurity highly, while role ambiguity and role conflict received low ratings. Further analysis will be conducted to clarify the significance of different job stressor results for the different demographics of the construction workers in the next section.

### 4.2. Reliability and Validity Tests

Confirmatory factor analysis (CFA) was conducted to test the fit of the measurement model to the data, and several coefficients were used to verify the fitness of the model, including $\chi^{2}/df$, IFI (Incremental Fitness Index), TLI (Tucker-Lewis index), CFI (Comparative Fitness Index) and RMSEA (Root Mean Square Error of Approximation). As presented in Table 3, the criteria showed a reasonably good fit for the measurement model to the data compared with statistical standards. IFI, TLI and CFI were greater than 0.9 and close to 1. Also, RMSEA was less than 0.08, which indicated that the model fits well.

The effectiveness of the questionnaires was tested by measuring their reliability and validity. Table 4 shows the results for the reliability of each dimension of the questionnaire. For reliability, Cronbach’s coefficient was applied and the results show that all the variables had good overall reliabilities (Cronbach’s alpha > 0.7), and were suitable for further data analysis [73]. For all variables, the SFL (standardized factor loading) estimates were higher than 0.5 and all the CR (composite reliability) values are greater than 0.7. Almost all the AVE values were higher than 0.5 except for pandemic fear, while its SFL estimate was generally 0.6 (>0.5) and the CR value was 0.81 (>0.7), and thus pandemic fear can also be acceptable [74]. These results indicate that the data of this measurement scale has good aggregation validity [70]. Table 5 shows that the square root of AVE for a dimension is greater than the largest correlation between the dimension and another dimension, thereby confirming the discriminant validity of the questionnaire.

## 5. Subgroup Analysis

### 5.1. Independent Sample T-Test

The differences in gender and marital status for COVID-induced and work-induced stressors among construction workers were verified by independent sample *t*-test, which is usually used to analyze the subgroup difference between two groups with nested demographic factors [75]. Table 6 shows the *t*-test results of the significant differences of perceived stressors for male and female workers, and Table 7 shows the *t*-test results for married and unmarried workers. It is worthy to note that this research selected the significance level of *p* < 0.1 as the acceptable level considering the limited number of samples [76]. The results show that only significant subgroup differences for work overload were found between the genders (*p* < 0.01). Particularly, males are more likely to perceive higher work overload than females. Hypothesis H1 proposed that gender differences would predict perceived stressors. Thus, hypotheses H1 is partially supported. It also found that, compared with unmarried workers, married workers perceive a relatively higher threat of being infected (*p* < 0.1), are more worried about their job security (*p* < 0.05), and are more sensitive to the poor working environment (*p* < 0.05), which can lead to higher job stress. Thus, hypothesis H2 is partially supported. 

### 5.2. ANOVA

ANOVA was applied to analyze the differences between groups in age, education level and working experience. Table 8 shows that significant subgroup differences existed between organizational pandemic response (*p* < 0.05), job insecurity (*p* < 0.001), role overload (*p* < 0.01) and role ambiguity (*p* < 0.05) for workers with different education levels. Therefore, hypothesis H3 is partially supported. Specifically, by comparing the mean values for different education levels, this indicates that highly-educated workers are more satisfied with the organizational responses to COVID-19 and have less anxiety about job security and are much clearer about their work roles, while they would perceive a higher workload during the pandemic. Table 9 shows that the degree of pandemic fear (*p* < 0.05) and job insecurity (*p* < 0.1) varied with the length of working experience. Experienced workers tend to have a lower fear of the COVID-19 pandemic and job insecurity. Therefore, hypothesis H4 is partially supported. From results of the ANOVA analysis of age in Table 10, it indicates that construction workers of different ages have different concerns about job security (*p* < 0.1). The results show that older construction workers, especially those aged 51 to 60, are more worried about their job security. Therefore, hypothesis H5 is partially supported.

## 6. Discussion

This study examined the influences of the demographic variables, including gender, marital status, age, working environment, and education level on the stressors perceived among construction workers during the current pandemic. The results confirmed that demographic characteristics could indeed lead to various distinctions on perceived stressors for construction workers. To the best of our knowledge, this study is one of the few to empirically explore the demographic influences of construction workers under the COVID-19 pandemic. It can provide meaningful insights for managers in promoting the health and well-being of construction workers during the COVID-19 pandemic.

### 6.1. Main Findings

The research revealed a significant relationship between gender and role overload. Male construction workers perceive higher workloads than the female workers. This is consistent with the results of Loosemore and Waters (2004) [57]. Construction is still a male-dominant industry, and female workers exist in a subordinate position, which means that they are often assigned support or assistance roles and perform relatively less technical tasks (e.g., handling of heavy equipment or tools) [77,78].

This research found that marital status had significant associations with certain sources of job stress. Married workers tend to perceive more fear of infection and threat of job security during the pandemic, and be more worried about the poor working environment compared with unmarried workers. This could be explained by the fact that married workers not only care about themselves, but there is also a greater sense of responsibility and concern for the well-being of their family members, especially the children and aging parents [51]. In addition, the possible project shutdown caused by the COVID-19 pandemic may have made them more worried about job stability and job prospects. Thus, they often have more financial pressure and can hardly bear the negative consequences of losing jobs. In addition, married workers tend to be risk-averse, and thus are more concerned about the exposure to the hazardous working environment onsite [79].

Education level is another critical demographic factor that can influence workers’ perceived stressors during the pandemic. Highly-educated workers are more inclined to be satisfied with organizational responses towards COVID-19 and less worried about their job insecurity. A good educational background means that these workers could better learn and understand the pandemic prevention measures. They tend to have positive attitudes towards making extra efforts to comply with pandemic prevention measures rather than regarding them as barriers towards production [80]. Due to limited social and financial resources, it is difficult for workers with less education to cope with the consequences of unemployment [81,82]. Therefore, these workers tend to pay more attention to the job insecurity during the pandemic. Furthermore, this research also revealed that a higher educational background could reduce workers’ ambiguity about work roles. Highly-educated workers are more aware of the complexity of construction tasks and understand their work roles much better. In addition, these workers reported higher workloads, meaning that they would undertake more tasks and responsibilities onsite.

The analysis also revealed statistically significant working experience differences in pandemic fear and job insecurity. Previous studies have found that the increased self-confidence among experienced workers could be protective against fear and uncertainty about infection risks and job stability induced by the pandemic [83]. By contrast, new workers tend to be cautious and alert to potential threats because they are unfamiliar with the working environment and have limited resources [84]. Furthermore, the results showed that age had a significant effect on workers’ perceived job insecurity. To some extent, this could be explained by the fact that older workers have poorer physical capabilities and less of a competitive advantage, which is associated with the lack of job stability [85]. In many regions in China, the contractors have been banned from employing male workers who are above 60 years of age, and female workers who are more than 50 years of age, which would further cause a sense of crisis for older workers [86].

Finally, regarding the overall mean scores of perceived stressors, this research found that construction workers perceived a slight threat of infection, job insecurity and excessive workload considering that mean scores of these stressors were close to or exceeded the midpoint of the corresponding score ranges. It seems to be understandable for the widespread fear and anxiety among construction workers due to the evolving nature and inherent scientific uncertainties of the COVID-19 pandemic, as well as the public countermeasures to prevent this virus (e.g., quarantine, isolation and physical distancing) [87]. In addition, the pandemic also led to increased workloads due to shortages in the workforce and the schedule pressure caused by the shutdowns and delays of construction projects at the early stages of the pandemic. In contrast, construction workers reported relatively higher satisfaction with organizational pandemic responses, the score of which is close to the high side of the score ranges. This can be attributed to the fact that China has implemented strict pandemic prevention policies, and the government has also issued specific guidance for construction projects [88].

### 6.2. Theoretical and Practical Implications

The findings of this research have theoretical implications for both pandemic prevention and occupational health research. Although the current COVID-19 pandemic has received a great deal of attention, few studies have focused on pandemic status in the context of the construction industry. This research should be one of the first comprehensive studies to examine the demographic influences on perceived stressors during the pandemic from the perspective of construction workers. The findings can provide a better understanding of how selected demographic characteristics (i.e., gender, marital status, age, education level and working experience) influence how construction workers perceive both COVID-induced and work-induced stressors. This can also give some meaningful insights for further research on the impact of the emerging COVID-induced stressors on the occupational health and safety of construction workers. 

This research is also anticipated to provide needed information for practitioners interested in promoting construction workers’ health and well-being through mitigating the job stressors during and after the pandemic. The findings indicate that the job stressors experienced by different demographic groups were unbalanced, causing different degrees of tension among the construction workers. Accordingly, management could design specialized health measures for the targeted demographic groups. First, flexible schedules and technical assistance equipment were suggested to reduce excessive workloads among male workers and highly-educated workers. Furthermore, management should care about the concerns of married workers and workers with low education levels with regard to infectious risks and job instability by necessary COVID-19 health education as well as some regular professional training activities. Moderately increased salary levels are also recommended for workers during the pandemic, which can reduce their concerns about job stability and stimulate their enthusiasm for work. Third, management is supposed to enhance the communication with workers with low education levels to ensure that they have a clear understanding about their work roles. Finally, management should improve the physical working environment through hazard identification and the provision of adequate safety equipment, which is expected to reduce the anxiety of married workers about over-exposure to occupational health and safety hazards onsite.

## 7. Conclusions

This research aims to empirically analyze the influence of demographic characteristics on the perception of occupational stressors among construction workers during the pandemic. The findings indicated that male workers experienced greater workloads than female workers. Married workers reported higher fear of infection, job insecurity and were more concerned about the poor working environment. Highly educated workers tend to be more satisfied with organizational pandemic responses, perceive lower job insecurity and role ambiguity, but perceive their workloads as being higher. The differences in pandemic fear and job insecurity were statistically significant with regards to working experience. The age difference was also significantly related to perceived job insecurity. This research could provide a comprehensive understanding on the effects of demographic differences on perceived stressors. It would provide meaningful insights for the project managers in promoting health and well-being among different demographic groups during and after the pandemic. 

Although useful findings were derived from the current study, the limitations should be clearly recognized. First, the sample involved in the current research was representative because the demographic information of workers was consistent with the general nature of Chinese construction workers. However, the data were merely collected from 21 projects in three Chinese provinces, and thus the generality of the results requires testing using a much larger sample in the future. Second, only five demographic factors were investigated in this research, and the effects of other demographic characteristics of construction workers such as the trade types, family economic status, and personal health status should be investigated further. 

## Figures and Tables

**Table 1 ijerph-19-04192-t001:** The demographic characteristics of respondents involved in this research and the comparison with other studies.

Characteristics	Category	Comparison of Sample Distribution between Different Studies			
		Sample of This Research: 498 Workers from 21 Chinese Construction Sites	Sample of Ye et al. (2022) [67]: 335 Workers from 7 Chinese Construction Sites	Sample of He et al. (2019) [68]: 536 Workers from 22 Chinese Construction Sites	Sample of Xie et al. (2022) [69]:172 Workers from 8 Chinese Construction Sites
Gender	Male	81.7%	87.5%	91.6%	91.3%
	Female	18.3%	12.5%	8.4%	8.1%
Marital status	Unmarried	17.3%	-	-	15.1%
	Married	82.7%	-	-	82.0%
Age	<30	17.5%	21.8%	20.9%	17.4%
	31–40	32.7%	23%	39.7%	32.6%
	41–50	41.4%	45.7%	28.4%	33.7%
	>50	8.4%	9.6%	11.0%	16.3%
Eudcation level	Primary school	15.7%	31.9%	17.9%	-
	Junior high school	43.6%	41.2%	44.3%	-
	Senior high school	25.5%	10.4%	18.0%	-
	Vocational college and above	15.2%	16.5%	19.8%	-
Experience	5<	29.5%	21.5%	27.0%	20.9%
	6–10	31.5%	35.5%	36.0%	26.7%
	10–15	21.1%	19.4%	22.2%	21.5%
	>15	17.9%	23.6%	14.8%	30.2%

**Table 2 ijerph-19-04192-t002:** Descriptive statistics of variables.

Constructs	M	SD	Skewness (Std.Error)	Kurtosis (Std.Error)
Pandemic fear	2.99	0.75	−0.20 (0.11)	−0.12 (0.22)
Organization response to COVID-19	3.72	0.86	−0.71 (0.11)	0.43 (0.22)
Job insecurity	3.22	0.89	−0.00 (0.11)	−0.44 (0.22)
Role overload	3.20	0.84	−0.15 (0.11)	−0.19 (0.22)
Role ambiguity	2.54	0.83	0.43 (0.11)	−0.20 (0.22)
Role conflict	2.63	0.83	0.33 (0.11)	−0.03 (0.22)
Poor working environment	2.86	0.91	0.22 (0.11)	−0.30 (0.22)

Note: Abbreviations: M = Mean value; SD = Standard deviation; Std.Error = standard error.

**Table 3 ijerph-19-04192-t003:** Results of CFA for the questionnaire.

Fitness Indices	$\chi^{2}/df$	IFI	TLI	CFI	RMSEA
Results	2.849	0.937	0.926	0.937	0.061
Standards	<3	>0.9	>0.9	>0.9	<0.08

Note: IFI = Incremental Fitness Index; TLI = Tucker-Lewis index; CFI = Comparative Fitness Index; RMSEA = Root Mean Square Error of Approximation.

**Table 4 ijerph-19-04192-t004:** Results of reliability and convergent validity for variables.

Constructs	Items	SFL	CR	AVE	Cronbach’s Alpha
Pandemic fear	PF1	0.65	0.78	0.43	0.79
	PF2	0.71			
	PF3	0.70			
	PF4	0.58			
	PF5	0.61			
Organizational pandemic responses	OPR1	0.79	0.90	0.74	0.90
	OPR2	0.94			
	OPR3	0.85			
Job insecurity	JI1	0.81	0.89	0.74	0.89
	JI2	0.92			
	JI3	0.84			
Role overload	RO1	0.82	0.88	0.71	0.88
	RO2	0.89			
	RO3	0.81			
Role ambiguity	RA1	0.80	0.89	0.66	0.89
	RA2	0.85			
	RA3	0.79			
	RA4	0.82			
Role conflict	RC1	0.76	0.87	0.63	0.87
	RC2	0.85			
	RC3	0.79			
	RC4	0.78			
Poor working environment	PWE1	0.79	0.91	0.73	0.91
	PWE2	0.88			
	PWE3	0.90			
	PWE4	0.84			

Note: Abbreviations: SFL = Standardized factor loading; CR = Composite reliability; AVE = Average variance extracted.

**Table 5 ijerph-19-04192-t005:** Results of discriminant validity.

No.	Constructs	1	2	3	4	5	6	7
1.	Pandemic fear	**0.65**						
2.	Organizational pandemic responses	0.15 **	**0.86**					
3.	Job insecurity	0.52 ***	0.31 ***	**0.86**				
4.	Role overload	0.37 ***	0.23 ***	0.36 ***	**0.84**			
5.	Role ambiguity	0.40 ***	−0.19 ***	0.30 ***	0.36 ***	**0.82**		
6.	Role conflict	0.44 ***	−0.19 ***	0.34 ***	0.46 ***	0.79 ***	**0.80**	
7.	Poor working environment	0.25 ***	−0.07 (n.s.)	0.37 ***	0.36 ***	0.38 ***	0.46 ***	**0.73**

Note: (1) Correlations are below the diagonal, and the figures in bold on the diagonal are the square root of the AVE of associated constructs. (2) *** *p* < 0.001, ** *p* < 0.01, n.s. *p* > 0.05.

**Table 6 ijerph-19-04192-t006:** Independent sample *t*-test of the gender differences for perceived stressors.

Constructs	Gender	M	SD	*t*-Value	*p*-Value
Pandemic fear	Male	2.99	0.77	0.191	0.849
	Female	2.97	0.65		
Organizational pandemic responses	Male	3.73	0.87	0.497	0.620
	Female	3.68	0.80		
Job insecurity	Male	3.23	0.88	0.771	0.442
	Female	3.15	0.91		
Role overload	Male	3.26	0.85	3.202	0.002
	Female	2.96	0.78		
Role ambiguity	Male	2.55	0.84	0.743	0.459
	Female	2.48	0.80		
Role conflict	Male	2.64	0.82	0.256	0.798
	Female	2.61	0.85		
Poor working environment	Male	2.88	0.92	1.080	0.282
	Female	2.77	0.90		

Note: M = Mean value; SD = Standard deviation.

**Table 7 ijerph-19-04192-t007:** Independent sample *t*-test of the marriage differences for perceived stressors.

Constructs	Marital Status	M	SD	*t*-Value	*p*-Value
Pandemic fear	Unmarried	2.85	0.70	−1.955	0.053
	Married	3.02	0.75		
Organizational pandemic responses	Unmarried	3.80	0.95	0.924	0.358
	Married	3.70	0.84		
Job insecurity	Unmarried	3.01	0.98	−2.208	0.029
	Married	3.26	0.86		
Role overload	Unmarried	3.12	0.91	−0.964	0.337
	Married	3.22	0.83		
Role ambiguity	Unmarried	2.41	0.82	−1.558	0.122
	Married	2.56	0.83		
Role conflict	Unmarried	2.56	0.84	−0.844	0.400
	Married	2.65	0.82		
Poor working environment	Unmarried	2.69	0.76	−2.215	0.028
	Married	2.89	0.94		

Note: M = Mean value; SD = Standard deviation.

**Table 8 ijerph-19-04192-t008:** ANOVA analysis of education levels.

Constructs	Education Level (M)					f-Value	*p*-Value
	Primary School	Junior High School	Senior High School	Vocational College	Bachelor Degree or Above		
Pandemic fear	2.92	3.03	3.01	2.94	2.78	0.697	0.626
Organizational pandemic responses	3.49	3.77	3.75	3.78	3.93	2.442	0.034
Job insecurity	3.21	3.39	3.11	3.11	2.56	5.063	0.000
Role overload	2.88	3.308	3.23	3.26	3.31	3.353	0.005
Role ambiguity	2.56	2.63	2.53	2.36	2.10	2.462	0.032
Role conflict	2.55	2.67	2.68	2.61	2.42	0.711	0.616
Poor working environment	2.71	2.92	2.88	2.88	2.76	0.795	0.554

Note: M = Mean value.

**Table 9 ijerph-19-04192-t009:** ANOVA analysis of working experience.

Constructs	Working Experience (M)					f-Value	*p*-Value
	≤2	3–5	6–10	10–15	>15		
Pandemic fear	3.26	2.83	3.03	3.06	2.95	2.686	0.031
Organizational pandemic responses	3.97	3.75	3.74	3.71	3.59	1.169	0.324
Job insecurity	3.49	3.04	3.26	3.28	3.20	2.101	0.080
Role overload	3.45	3.16	3.13	3.29	3.21	1.287	0.274
Role ambiguity	2.66	2.40	2.56	2.66	2.50	1.605	0.172
Role conflict	2.53	2.54	2.68	2.73	2.59	1.043	0.384
Poor working environment	2.91	2.69	2.97	2.90	2.81	1.740	0.140

Note: M = Mean value.

**Table 10 ijerph-19-04192-t010:** ANOVA analysis of age.

Constructs	Age (M)						f-Value	*p*-Value
	≤20	21–30	31–40	41–45	46–50	51–60		
Pandemic fear	3.25	2.89	2.94	3.03	3.01	3.16	0.863	0.522
Organizational pandemic responses	3.17	3.84	3.63	3.65	3.74	4.02	1.701	0.119
Job insecurity	3.33	3.03	3.13	3.29	3.32	3.52	1.989	0.066
Role overload	3.67	3.14	3.22	3.17	3.25	3.27	0.530	0.786
Role ambiguity	3.00	2.47	2.55	2.60	2.55	2.42	0.595	0.734
Role conflict	3.13	2.65	2.66	2.66	2.59	2.46	0.580	0.746
Poor working environment	3	2.72	2.85	2.90	2.83	3.13	0.995	0.428

Note: M = Mean value.

## Data Availability

The data presented in this study are available on request from the corresponding author.

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
