# Peer review of "Demographic Influences on Perceived Stressors of Construction Workers during the COVID-19 Pandemic"

_ijerph, 2022, doi:10.3390/ijerph19074192_

Round 1

Reviewer 1 Report

The title and the content of this paper suggest that the authors investigated the stressors of construction workers during the COVID-19 pandemic.  It is however disappointing that the terminology used in the paper does not agree with the title and the content of the paper. In my opinion, the research described here relates to a survey of construction workers’ work-related stressors, one of which happens to be Covid-related stressors.  So, the authors should refrain from mentioning “Covid-related stressors” and “work-related stressors” as two different types of stressors.  It would be more appropriate to state that the “work-related stressors” that are generally mentioned in the literature have been augmented in this study by “pandemic fear” a Covid-related stressor in the period when the world experienced a Covid pandemic.  Actually, the researchers investigated work-related stressors which inevitably include Covid-related stressors because of the Covid pandemic.  Had there been a plague, the work-related stressors would have been augmented by plague-related stressors in the study; in times of no pandemic, there would be no such stressors in the study.   So, I encourage the authors to replace the terminology “COVID-induced and work-induced stressors among construction workers” by “work-induced stressors among construction workers during the Covid pandemic”.

The sampling method that the researchers used to determine the original 800 individuals that were handed a questionnaire is not clear to me.  I suggest that the following two points be acknowledged in the paper as limitations of the study.

  • Did the researchers pick the 21 companies at random from the directory of an organization such as a Contractors Association? Did they approach 21 companies with which they had relationships in the past?  Did they pick these companies on the recommendations of local individuals that have a good knowledge of the industry in their region?  In sum, how did they pick these 21 construction companies.  The nature of the population at large is important in this type of research. 

Did they pick the workers in these companies to fit a certain trade profile (e.g., steel workers, bricklayers, unskilled labor, etc.)?  We know the respondents’ gender, age, education, marital status, and industry experience, but we are missing information about their trades.  This information could be as important as (in my opinion, more important than) gender, age, education, marital status, and industry experience.  If one does not know the nature of the larger population, the nature of the respondents, and the sampling methods used, it is difficult to assess the value of the responses.  

Author Response

Responses to Referees’ Comments

Referee #1

  • Referee’s comment one:

Referee #1: The title and the content of this paper suggest that the authors investigated the stressors of construction workers during the COVID-19 pandemic.  It is however disappointing that the terminology used in the paper does not agree with the title and the content of the paper. In my opinion, the research described here relates to a survey of construction workers’ work-related stressors, one of which happens to be Covid-related stressors.  So, the authors should refrain from mentioning “Covid-related stressors” and “work-related stressors” as two different types of stressors.  It would be more appropriate to state that the “work-related stressors” that are generally mentioned in the literature have been augmented in this study by “pandemic fear” a Covid-related stressor in the period when the world experienced a Covid pandemic.  Actually, the researchers investigated work-related stressors which inevitably include Covid-related stressors because of the Covid pandemic.  Had there been a plague, the work-related stressors would have been augmented by plague-related stressors in the study; in times of no pandemic, there would be no such stressors in the study.   So, I encourage the authors to replace the terminology “COVID-induced and work-induced stressors among construction workers” by “work-induced stressors among construction workers during the Covid pandemic”.

Response to comment:

Thank you very much for your kind suggestion. The revision version did not distinguish COVID-induced from Work-induced stressors. We have relevant revision throughout the manuscript to avoid unnecessary misunderstanding.

  • Referee’s comment two:

Referee #1: Did the researchers pick the 21 companies at random from the directory of an organization such as a Contractors Association? Did they approach 21 companies with which they had relationships in the past?  Did they pick these companies on the recommendations of local individuals that have a good knowledge of the industry in their region?  In sum, how did they pick these 21 construction companies. The nature of the population at large is important in this type of research. 

Response to comment:

These projects were selected based on the previous cooperation relationships as well as the permission of the corporate level managers, and 30-50 workers were randomly recruited from each project. To confirm the representativeness of the sample involved in this research, we have compared the demographic information between this research and other three studies conducted in Chinese construction industry recently. It found that the demographic distribution in this research was consistent with the general nature of Chinese construction workers identified in other studies. Nevertheless, we still acknowledged the limitation of generalization of findings at the end to guide the future research direction. The revision was located in Page 6, “3.2 Questionnaire survey” and Page 13 “7. Conclusion”.  

  • Referee’s comment three:

    Referee #1: Did they pick the workers in these companies to fit a certain trade profile (e.g., steel workers, bricklayers, unskilled labor, etc.)?  We know the respondents’ gender, age, education, marital status, and industry experience, but we are missing information about their trades.  This information could be as important as (in my opinion, more important than) gender, age, education, marital status, and industry experience.  If one does not know the nature of the larger population, the nature of the respondents, and the sampling methods used, it is difficult to assess the value of the responses.  

Response to comment:

Thanks a lot for your comment. We agree with your opinion, but the information on trade type was not included in this investigation. We have acknowledged the limitation at the end, and other meaningful demographic factors such as family economic status and trade types could be explored in the future research. The comparison between the sample of this research and that in other studies on Chinese construction workers recently. It indicated that the sample involved in this research is to some extent consistent with the general characteristics of Chinese construction workers. The revision was located in Page 6, “3.2 Questionnaire survey” and Page 13 “7. Conclusion”.

Reviewer 2 Report

The authors identify the relationship between demographic characteristics of construction workers and professionals and the perceived impact of COVID-19. While the writing is good, the manuscript must be improved to show a better problem statement, research gap, and research design. Some suggestions have been provided that might help address that concern (based on the previous round of review).

1 - Thank you for revising the manuscript while considering my suggestions. Below are my responses as per the responses received.

2 - Thank you for adding the data analysis approach. Please consider adding the data collection approach too.

3 - Sorry, I cannot identify the newly added theoretical contribution. Suggest adding the line number or highlighting it during the next round of review. Maybe can also add something like "this research contributes to the body of knowledge by...." in one of the paragraphs in the introduction.

4 - I agree that the manuscript is on demographic influences on workers' perceived stressors (during the COVID-19). In other words, the study is identifying the impact of COVID-19 on worker stress. Also, I believe the study should be proposing solutions or strategies to reduce COVID-19 impacts on worker stress. Therefore, there is a need to synthesize the current literature on COVID-19 impacts and strategies to show the research gap that the manuscript is trying to fill in. Alternatively, the manuscript can also consider synthesizing what others have done on identifying the relationship between demographic influences and worker stress, to show an alternative research gap. In summary, these are all suggestions to help the manuscript show the research gap that the study is trying to fill.

5 - Thank you for the detailed explanation. Please consider adding them to the manuscript as well as the necessary references to back up those statements.

6 - Thank you. However, I still do not understand why the study adopts the survey from Chi et al. 2020, Tang et al. 2022, and so on. Why not other studies? Please add justifications and the steps taken in selecting an appropriate survey, so others can repeat the methodological process to ensure the study is replicable by others.

7 - Thank you for the explanation.

8 - Thank you for the explanation. However, that was the inclusion and exclusion criteria for filtering the data for data analysis. The question was more on what were the inclusion and exclusion criteria for selecting the survey respondents? For example, what are the required profession, level of education, work location, number of experiences to be able to participate in the survey? Or anyone can participate freely?

9 - Thank you for considering the suggestion.

10 - Thank you for the detailed explanation. Please consider adding them to the manuscript as well as the necessary references to back up those statements.

11- Thank you for considering the suggestion.

12 - Thank you for the explanation. However, while the data is normally distributed, I believe ANOVA is not the appropriate data analysis technique to analyze Likert-scale data as ANOVA is more appropriate for continuous data while Likert-scale is ordinal data. Hope you could verify accordingly.

13 - Thank you for considering the suggestion.

Author Response

Responses to Referees’ Comments

Referee #2

  • Referee’s comment one:

Referee #2: The authors identify the relationship between demographic characteristics of construction workers and professionals and the perceived impact of COVID-19. While the writing is good, the manuscript must be improved to show a better problem statement, research gap, and research design. Some suggestions have been provided that might help address that concern (based on the previous round of review).

1 - Thank you for revising the manuscript while considering my suggestions. Below are my responses as per the responses received.

2 - Thank you for adding the data analysis approach. Please consider adding the data collection approach too.

Response to comment:

Thank you very much for your comments. We have revised the manuscript accordingly and we hope the revision could meet your requirement. The data collection approach has been added in the section of Abstract.

  • Referee’s comment two:

Referee #2: Sorry, I cannot identify the newly added theoretical contribution. Suggest adding the line number or highlighting it during the next round of review. Maybe can also add something like "this research contributes to the body of knowledge by...." in one of the paragraphs in the introduction.

Response to comment:

Thank you very much for your comments. We have added the theoretical contribution in the section of Introduction. In practice, the theoretical implications have been emphasized in the Section of Discussion. The newly added content has been highlighted in the revision version.

  • Referee’s comment three:

Referee #2: I agree that the manuscript is on demographic influences on workers' perceived stressors (during the COVID-19). In other words, the study is identifying the impact of COVID-19 on worker stress. Also, I believe the study should be proposing solutions or strategies to reduce COVID-19 impacts on worker stress. Therefore, there is a need to synthesize the current literature on COVID-19 impacts and strategies to show the research gap that the manuscript is trying to fill in. Alternatively, the manuscript can also consider synthesizing what others have done on identifying the relationship between demographic influences and worker stress, to show an alternative research gap. In summary, these are all suggestions to help the manuscript show the research gap that the study is trying to fill.

Response to comment:

Thank you very much for your kind recommendation. We have rewritten the Introduction section and synthesized the current literature on COVID-19-studies in construction industry, as well as demographic research about construction workers, based on which the research gaps were proposed much logically. The revision was located in the section of Introduction.

  • Referee’s comment four:

Referee #2: 5 - Thank you for the detailed explanation. Please consider adding them to the manuscript as well as the necessary references to back up those statements.

Response to comment:

Thank you very much for your comments. We have added all necessary references accordingly.

  • Referee’s comment five:

Referee #2: However, I still do not understand why the study adopts the survey from Chi et al. 2020, Tang et al. 2022, and so on. Why not other studies? Please add justifications and the steps taken in selecting an appropriate survey, so others can repeat the methodological process to ensure the study is replicable by others.

Response to comment:

Thanks a lot for your comments. Structured survey questions of all constructs were adapted from self-report items validated by previous studies. Borrowing scales from other studies is quite common in empirical studies based on questionnaire. In this research, we firstly selected the widely-used scales for each construct from previous studies and these scales should have been validated in similar context such as the occupational health management in the organization or the current COVID-19 pandemic. Then we would request the scales from the authors of these studies if the scales were not presented in their published papers. Based on the normal step of empirical studies, the reliability and validity of the questionnaires should be confirmed before they were used to test the hypotheses. We believe that the scales used in this research should be appropriate to capture the data on perceived stressors among the construction workers during the pandemic. The revision was located in Page 5, “3.1 measures”.

  • Referee’s comment six:

Referee #2: However, that was the inclusion and exclusion criteria for filtering the data for data analysis. The question was more on what were the inclusion and exclusion criteria for selecting the survey respondents? For example, what are the required profession, level of education, work location, number of experiences to be able to participate in the survey? Or anyone can participate freely?

Response to comment:

        To make the sample much representative for the larger population of Chinese construction workers, 30-50 frontline workers were randomly recruited without any criteria from each project. All workers on site were informed of the questionnaire by the management, and they could participated voluntarily.

  • Referee’s comment seven:

    Referee #2: 10 - Thank you for the detailed explanation. Please consider adding them to the manuscript as well as the necessary references to back up those statements.

Response to comment:

        Thank you very much for your comments. We have added all necessary references accordingly.

  • Referee’s comment eight:

Referee #2: However, while the data is normally distributed, I believe ANOVA is not the appropriate data analysis technique to analyze Likert-scale data as ANOVA is more appropriate for continuous data while Likert-scale is ordinal data. Hope you could verify accordingly.

Response to comment:

        Thanks a lot for your comment. According to Boone and Boone (2012)[1], although the numbers assigned to Likert-type items that express a “greater than” relationship fall into the ordinal data, the Likert-scale data are created by calculating a composite score (sum or mean) from several Likert-type items can be analyzed by ANOVA. In this research, the score of one stressor were determined by calculating the mean values of all the corresponding questions. For instance, pandemic fear of each worker is the mean value of the scores of its five specific questions. Therefore, ANOVA is suitable for this research. In practice, we found that a huge amount of previous studies have used the ANOVA to carry out the data analysis of Likert scale questions in previous studies (e..g, Meng and Chan (2020)[2], Jin et al. (2017)[3], Han et al. (2019)[4]).

  1. Boone, H.N.; Boone, D.A. Analyzing likert data. Journal of extension 2012, 50, 1-5.
  2. Meng, X.; Chan, A.H. Demographic influences on safety consciousness and safety citizenship behavior of construction workers. Saf. Sci. 2020, 129, 104835.
  3. Jin, R.; Hancock, C.M.; Tang, L.; Wanatowski, D. Bim investment, returns, and risks in china’s aec industries. J. Constr. Eng. Manage. 2017, 143, 04017089.
  4. Han, Y.; Feng, Z.; Zhang, J.; Jin, R.; Aboagye-Nimo, E. Employees’ safety perceptions of site hazard and accident scenes. J. Constr. Eng. Manage. 2019, 145, 04018117.

Round 2

Reviewer 2 Report

The revised manuscript has addressed all feedback comprehensively. Therefore, I have no further feedback to add from my side.

This manuscript is a resubmission of an earlier submission. The following is a list of the peer review reports and author responses from that submission.

Round 1

Reviewer 1 Report

The title of this paper suggests that the authors investigated the stressors of construction workers during the COVID-19 pandemic.  It is disappointing that the contents of the paper are not related to the title as the research described here relates to a survey of construction workers’ COVID-19-related stressors, and separately from this, workers’ work-related stressors.  The idea to look into the effect of five demographic characteristics (gender, marital status, education, work experience, and age) on stressors is good, but in the absence of a comparison of the work stressors during the pandemic and before the pandemic, it is impossible to say that what the authors recorded during the pandemic is any different than what was recorded during a pandemic-free time.  As such, I doubt that there is any value in the results of the survey presented in this paper.

The English of the paper is quite poor and needs to be radically edited.  Awkward sentence structure, awkward terms, and awkward idioms abound in this paper and are very disturbing to the reader.  I strongly recommend that the authors totally rewrite this paper if they decide to submit it again to this or any other journal.

The information presented in the paper is quite sloppy, in that what is mentioned in one part of the paper conflicts in another part of the paper.  Also, quite a bit of vital information is missing. 

  • Line 194: It is stated that five questions were used to measure “pandemic fear”, and yet according to Table 3, there are six questions. Which one is correct?  Out of the five (or is it six?) questions, only two are given in the paper and the remaining three (or is it four?) questions are unknown to the reader.  Where are the other questions?  When the entire research is based on these questions, it is not acceptable for the authors to mention only a sample of two questions.
  • Line 199: It is stated that four questions were used to measure “job insecurity”, and yet according to Table 3, there are three questions. Which one is correct?  Out of the four (or is it three?) questions, only two of them are given in the paper and the remaining two (or is it one?) questions are unknown to the reader.  Where are the other questions?  When the entire research is based on these questions, it is not acceptable for the authors to mention only a sample of two questions.
  • Line 205: The number of questions that were used to measure “organizational pandemic responses” is not stated but we understand from Table 3 that there are three questions. Out of the three questions, only two of them are given in the paper and the remaining one question is unknown to the reader.  Where is the other question?  When the entire research is based on these questions, it is not acceptable for the authors to mention only a sample of two questions.
  • Line 211: It is stated that four questions were used to measure “role overload” and yet according to Table 3, there are three questions. Which one is correct?  Out of the four (or is it three?) questions, only two of them are given in the paper and the remaining two (or is it one?) questions are unknown to the reader.  Where are the other questions?  When the entire research is based on these questions, it is not acceptable for the authors to mention only a sample of two questions.
  • Line 215: It is stated that four questions were used to measure “role ambiguity”. Out of the four questions, only two of them are given in the paper and the remaining two questions are unknown to the reader.  Where are the other questions?  When the entire research is based on these questions, it is not acceptable for the authors to mention only a sample of two questions.
  • Line 219: It is stated that three questions were used to measure “role conflict” and yet according to Table 3, there are four questions. Which one is correct?  Out of the three (or is it four?) questions, only two of them are given in the paper and the remaining one (or is it two?) questions are unknown to the reader.  Where are the other questions?  When the entire research is based on these questions, it is not acceptable for the authors to mention only a sample of two questions.
  • Line 223: It is stated that four questions were used to measure “poor working environment”. Out of the four questions, only two of them are given in the paper and the remaining two questions are unknown to the reader.  Where are the other questions?  When the entire research is based on these questions, it is not acceptable for the authors to mention only a sample of two questions.

The authors state that 498 valid questionnaires were received from the respondents.  What is not clear to me is whether these 498 respondents represent all the workers in the three provinces that the authors targeted?  Is so, what were the sampling techniques they used to determine the original sample of the population at large that were sent questionnaires.  Also, what was the rate of response?  If one does not know the nature of the larger population, the sampling methods used, and the rate of response, it is impossible to assess the value of the responses.  It is certainly impossible to generalize the findings.

I am not commenting on the findings, the discussion and the implications of the findings, and the contribution of the findings to the general knowledge about job stressors in construction sites because I am not satisfied with the method used in the study, and with the sloppiness in the paper. 

Reviewer 2 Report

The authors identify the relationship between demographic characteristics of construction workers and professionals and the perceived impact of COVID-19. While the writing is good, the manuscript must be improved to show a better problem statement, research gap, research design and discussion. Some suggestions have been provided that might help address that concern.

Abstract: Please include a brief methodology in the abstract.

Introduction: The problem statement of the research is not explained well, making me question why do we need to know about the relationship between construction worker demographics and the impacts of COVID-19? Understanding the context where understanding the relationship is very important. The introduction section will benefit from re-writing to articulate the motivation for the research by explaining the need of understanding that relationship.

Literature review: What have others have done related to identifying the impacts and pandemic responses? Suggest adding a subsection on both items. A simple search on the topic shows the following papers exist. Please consider reviewing and others them accordingly:

Impacts

Agyekum, K., Kukah, A. S., & Amudjie, J. (2021). The impact of COVID-19 on the construction industry in Ghana: the case of some selected firms. Journal of Engineering, Design and Technology.

Al-Mhdawi, M. K. S., Brito, M. P., Abdul Nabi, M., El-Adaway, I. H., & Onggo, B. S. (2022). Capturing the impact of COVID-19 on construction projects in developing countries: A case study of Iraq. Journal of Management in Engineering38(1), 05021015.

Ogunnusi, M., Omotayo, T., Hamma-Adama, M., Awuzie, B. O., & Egbelakin, T. (2021). Lessons learned from the impact of COVID-19 on the global construction industry. Journal of engineering, design and technology.

King, S. S., Rahman, R. A., Fauzi, M. A., & Haron, A. T. (2021). Critical analysis of pandemic impact on AEC organizations: The COVID-19 case. Journal of Engineering, Design and Technology.

Jeon, J., Padhye, S., Bhattacharyya, A., Cai, H., & Hastak, M. (2022). Impact of COVID-19 on the US Construction Industry as Revealed in the Purdue Index for Construction. Journal of Management in Engineering38(1), 04021082.

Rehman, M. S. U., Shafiq, M. T., & Afzal, M. (2021). Impact of COVID-19 on project performance in the UAE construction industry. Journal of Engineering, Design and Technology.

Strategies

Assaad, R., & El-adaway, I. H. (2021). Guidelines for responding to COVID-19 pandemic: Best practices, impacts, and future research directions. Journal of Management in Engineering37(3), 06021001.

Zamani, S. H., Rahman, R. A., Fauzi, M. A., & Yusof, L. M. (2022). Government pandemic response strategies for AEC enterprises: lessons from COVID-19. Journal of Engineering, Design and Technology.

Salami, B. A., Ajayi, S. O., & Oyegoke, A. S. (2021). Coping with the COVID-19 pandemic: an exploration of the strategies adopted by construction firms. Journal of Engineering, Design and Technology.

Kim, S., Kong, M., Choi, J., Han, S., Baek, H., & Hong, T. (2021). Feasibility analysis of COVID-19 response guidelines at construction sites in south korea using CYCLONE in terms of cost and time. Journal of Management in Engineering37(5), 04021048.

Jones, W., Gibb, A. G., & Chow, V. (2021). Adapting to COVID-19 on construction sites: what are the lessons for long-term improvements in safety and worker effectiveness?. Journal of Engineering, Design and Technology.

Raoufi, M., & Fayek, A. R. (2022). New Modes of Operating for Construction Organizations during the COVID-19 Pandemic: Challenges, Actions, and Future Best Practices. Journal of Management in Engineering38(2), 04021091.

Methodology

Explain why questionnaire survey is suitable. The fact that other construction management studies have also employed a similar approach when developing surveys is not enough of a reason to adopt the same

How did the authors decide which survey to adopt (e.g., Chi et al. 2020, Tang et al. 2020)? Please consider explaining them in detail.

What was the process involved in developing the questions that were not adopted from prior studies? Also, how do the authors validate whether the survey is reliable or otherwise? Please explain in detail as if the survey is not reliable, the data can be questionable.

What were the inclusion and exclusion criteria for the study respondents? 

Please consider attaching the survey questionnaire as an appendix. That would help readers to understand better on the survey design.

How did the study determine the minimum target number of respondents? Can the 92 female respondents sufficient to represent the whole female population in the Chinese construction industry?

What are the variables used in the survey to represent each construct?

What is the distribution of the collected data? Is it normally distributed or otherwise? Also, is the selected analysis the appropriate analysis for Likert scale data or non-normally distributed data?

Discussion:

The authors need to compare and contrast their findings with those of previous studies. It is not enough to just report the findings of previous studies. The major problem of this paper is that the WHY question is not answered. WHY are the different respondent characteristics can influence their perceived impact of COVID-19? WHY are construction professionals and workers have different perceptions compared to other industries? 

Please describe the shortcomings of this research clearly and rewrite the conclusion to be concise and focused on the results and discussion of this research.

Reviewer 3 Report

The authors have conducted an interesting research topic, which focuses on the construction workers during the pandemic. 

The research methodology has been well described and elaborated. The authors have clearly presented the results and discussed robustly the outcomes of the present research. 

The way how the research implications are present is unquestionable. 

Very minor concern I have is just to make some improvements in the literature background in the introduction section.